# Diagnostics and Screening in Breast Cancer with Brain and Leptomeningeal Metastasis: A Review of the Literature

**DOI:** 10.3390/cancers16213686

**Published:** 2024-10-31

**Authors:** Adam J. Cohen-Nowak, Virginia B. Hill, Priya Kumthekar

**Affiliations:** 1Department of Internal Medicine, McGaw Medical Center of Northwestern University, Chicago, IL 60611, USA; adam.cohennowak@nm.org; 2Neuroradiology Section, Department of Radiology, Northwestern University Feinberg School of Medicine, Chicago, IL 60611, USA; virginia.hill@nm.org; 3Department of Neurology, Northwestern Memorial Hospital, Chicago, IL 60611, USA

**Keywords:** breast cancer, diagnostics, screening, brain metastasis, leptomeningeal carcinomatosis

## Abstract

Patients with advanced stages of breast cancer are at high risk of the cancer metastasizing or spreading to areas of the central nervous system, including the brain, cerebrospinal fluid, and the protective tissue layers of the brain, termed leptomeninges. Although guidelines recommend screening for central nervous system disease in other cancers, such as non-small cell lung cancer, there are no such recommendations for patients with breast cancer unless they have symptoms such as headaches, weakness, or vomiting. This review discusses the evidence behind screening for breast cancer that has spread to the central nervous system, as well as new methods of diagnosis, including specialized imaging, serum testing, and cerebrospinal fluid analysis.

## 1. Introduction

Brain and leptomeningeal metastasis remain a highly morbid complication of advanced breast cancer, with a major impact on quality of life and overall survival. According to a systematic review of breast cancer with brain metastasis (BCBM), patients had a median of about 15 months of survival after diagnosis of brain metastasis, with the most common presenting symptoms including headache, nausea, and hemiparesis [1]. Estimating survival after diagnosis of leptomeningeal (LM) disease is difficult, but after subgroup analysis controlling for pre-existing cranial metastasis, it is likely weeks to months [2]. However, current advances in cancer therapeutics, including immunotherapy, targeted therapy, surgery, radiotherapy, and systemic chemotherapy regimens, have improved life expectancy [3]. While the National Comprehensive Cancer Network (NCCN) guidelines include screening recommendations for brain and leptomeningeal disease in patients with non-small cell lung cancer (NSCLC) and certain hematologic malignancies, there are no such screening recommendations for breast cancer. Based on prior autopsy studies of patients with breast cancer, the rate of subclinical brain and LM metastasis is likely much higher than estimated, with about 30% of all cases demonstrating cranial and LM involvement in patients without symptoms of central nervous system (CNS) involvement [4]. This raises the question of whether to screen asymptomatic breast cancer patients for BCBLM with the possibility of earlier CNS-directed treatment, including changes to the selection of systemic therapy. This review will focus on the literature surrounding screening and diagnosis of brain and leptomeningeal disease in patients with breast cancer.

A literature search was conducted through the Northwestern University library, including but not limited to PubMed Central, EBSCO, Elsevier, Science Direct, and Wiley Online Library. Search terms included breast cancer, brain metastasis, leptomeningeal metastasis, breast cancer with brain metastasis (BCBM), diagnosis of BCBM, screening for BCBM, diagnosis of leptomeningeal metastasis, and leptomeningeal carcinomatosis. Studies were screened based on relevance. No studies were excluded based on the date of publication.

## 2. Screening

Routine screening for BCBM in asymptomatic patients is a controversial topic. Although breast cancer is the most common solid tumor to spread to the leptomeninges and the second-most common to spread to the brain, patients with high-risk breast cancers such as metastatic HER2+ or triple-negative breast cancer (TNBC) are not routinely screened for CNS involvement unless they have a suggestion of neurologic symptoms [5,6].

Several retrospective studies have looked at outcomes in patients with BCBM compared to other cancers such as lung and melanoma. At Dana-Farber, a retrospective study of 349 breast cancer patients found that BCBM patients were more likely to be symptomatic at presentation, experience seizures, harbor brainstem or leptomeningeal involvement, and experience brain death compared to patients with NSCLC [7]. Another study of 959 patients at Moffitt Cancer Center with melanoma, NSCLC, and breast cancer demonstrated that BCBM patients were more likely to be younger at presentation and experience worse overall survival (mean 9.9, 10.3, and 13.7 months, respectively, *p* = 0.0006) [8]. Altogether, these studies suggest we are detecting BCBM once patients are symptomatic with later stages of CNS disease, consistent with guideline-directed practices. Another study out of Japan evaluated outcomes in a population of 1256 patients with BCBM and found there were several factors associated with improved overall survival (OS), including diagnosis of brain metastasis within 6 months of metastatic disease, asymptomatic brain metastasis, and hormone-positive or HER2-positive breast cancer [9]. This raises the possibility of improved OS with earlier detection of BCBM in patients with metastatic disease and without symptoms of CNS involvement. About a third of patients in the study had TNBC (337/1256), and 75% had symptomatic CNS disease, with asymptomatic disease spread evenly among the subgroups. Time from primary diagnosis to brain metastasis and length of metastasis-free survival were longer for luminal-type tumors, including HER2-positive compared with TNBC. Interestingly, the authors reference an older prospective trial from 1998–2001 that employed screening CT Brain or MRI for BCBM and demonstrated similar survival between asymptomatic and symptomatic patients with BCBM (23/96 occult, 73/96 symptomatic) [10]. However, the authors explain this discrepancy in outcomes data with the arrival of new systemic therapies such as HER2-targeted agents, advanced diagnostic imaging, and new treatment modalities for CNS disease. Other studies have confirmed poorer outcomes for patients with TNBC or HER2-negative metastases [11,12,13]. Our institution performed a retrospective study and determined that about 6% of all patients (*N* = 1218) with Stage II-III breast cancer went on to develop brain metastases over a median of 92 months follow-up, with TNBC (HR 2.043), higher grade (HR = 1.667), and stage (HR = 3.851) all independent risk factors for earlier development of BCBM [14]. Overall survival for BCBM patients with TNBC is poor, with median OS reported anywhere from about 5 to 8 months [9,15].

Leptomeningeal disease (LMD) is a feared complication of advanced breast cancer with CNS metastasis. Likewise, there is no guideline for screening patients with advanced breast cancer for LMD with MRI Brain and/or lumbar puncture with cytology, compared with hematologic malignancies such as acute lymphoblastic leukemia and several B-cell lymphomas [16]. Among 153 patients at The Ohio State University Comprehensive Cancer Center diagnosed with LMD from 2010–2015, the majority were breast cancer patients (43%) followed by lung and melanoma, with Stage III-IV disease at presentation (71%) and a median OS of 1.9 months (95% CI: 1.3–2.5 months). Less than half (44%) who underwent diagnostic lumbar puncture for MRI evidence of LMD had positive cytology, which was associated with poorer OS [17]. Interestingly, HER2-positive BC appears less likely than luminal and TNBC to metastasize to the leptomeninges, which may have implications if screening for LMD is ever advised in advanced breast cancer [18]. There does not appear to be a survival benefit in HER2 expression for LMD [19].

If screening guidelines are to change for patients with breast cancer, the consequences are likely to be (1) an increased incidence and prevalence of metastatic breast cancer, specifically BCBLM diagnosis; (2) earlier CNS-directed therapies, including systemic therapy, radiation therapy, and invasive neurosurgical procedures; and (3) artificially improved length of survival for BCBLM patients. Whether this would have any impact on quality of life and OS is the crucial question. There are currently five clinical trials in process to screen asymptomatic patients with advanced breast cancer for CNS involvement: NCT03881605 (metastatic TNBC/HER2-positive), NCT03617341 (metastatic/unresectable HER2-positive), NCT00398437 (Stage IV HER2-positive), NCT05115474 (all Stage IV breast cancer), NCT06247449 (TNBC/HER2-positive Stage II or III), and NCT04030507 (TNBC, HR-positive, HER2-positive, inflammatory breast cancer). All studies plan to utilize MRI Brain with and without contrast for screening, with one utilizing a newer technology of chemical exchange saturation transfer (NCT03881605), which is discussed in the diagnostics section below. These studies will hopefully provide evidence as to whether screening for BCBLM is indicated in metastatic breast cancer. Currently, National Comprehensive Cancer Network (NCCN) guidelines state that an MRI Brain with contrast is recommended only for suspicious CNS symptoms in Stage IV (M1) or recurrent invasive breast cancer [20]. However, aside from patients with Stage IA NSCLC, all other NSCLC patients are recommended to have an MRI Brain with contrast to screen for CNS metastasis (IIA recommendation) [21]. These recommendations appear to be at least partly based on studies looking at cost–benefit analysis of screening MRI to limit unnecessary craniotomies in NSCLC patients with multiple metastases [22,23]. Further studies are needed to evaluate the cost–benefit analysis of screening MRI Brain for breast cancer patients. This is most likely to influence the NCCN to recommend screening in this population.

## 3. Diagnostics

The “gold standard” for diagnosing brain metastasis and leptomeningeal carcinomatosis is MRI Brain with gadolinium enhancement and lumbar puncture with cytology of cerebrospinal fluid, respectively. However, there are limitations in both the sensitivity and specificity for the detection of metastases in these technologies. Newer methods aim to detect metastatic CNS disease with more accuracy and to discriminate recurrence from post-surgical or radiation-related changes. These include more advanced contrasted imaging studies and laboratory analysis of detecting circulating tumor cells, genomic material, oncoproteins, and metabolites termed “liquid biopsy”, which will be highlighted in the following sections (Figure 1).

### 3.1. Imaging

While MRI Brain with gadolinium enhancement is the gold standard for the detection of BCBLM, there are still shortcomings in its diagnostic capabilities. Several studies have sought to further improve the diagnostic yield of MRI to aid in treatment planning and prognosticate metastatic disease. There are a variety of MRI pulse sequences and techniques that aid in the detection of metastatic disease, including diffusion-weighted imaging (DWI), fluid-attenuated inversion recovery (FLAIR), magnetic resonance spectroscopy (MRS), dynamic susceptibility contrast (DSC) and dynamic contrast-enhanced (DCE) perfusion-weighted imaging (PWI), and resting-state functional MRI (rsfMRI) [24]. Several clinical trials are underway to compare new methods in imaging to standard of care and to improve sensitivity in the detection of metastatic lesions (Table 1). While MRI is the gold standard, a trial at Memorial Sloan Kettering evaluated 18F-FLT-PET imaging in patients with BCBM treated with whole brain radiation therapy (WBRT) with or without sorafenib and found a steeper decline in SUVmax in the sorafenib group, noting a possible utility of PET imaging in this application [25].

One of the challenges in diagnostic imaging is discriminating brain metastasis from a second primary tumor. Zhang et al. used in-house-developed software to perform a machine learning textural analysis on apparent diffusion coefficient (ADC) images from the diffusion-weighted imaging sequence by discriminating tumors based on heterogeneity. They calculated metrics from an ADC-based histogram analysis and used gray-level co-occurrence matrix (GLCM) parameters to analyze ADC maps in 76 GBMs (glioblastomas) and 90 metastases, including three breast cancer metastases. Based on homogeneity, imaging was able to differentiate GBMs from metastases with an AUC of 0.886, sensitivity of 83.3%, and specificity of 76.9% [26]. Similarly, Skogen et al. performed texture analysis on diffusion-tensor images (with mapping of white matter tracts) of 22 GBMs and 21 metastases (5 breast cancer metastases) and found that heterogeneity was increased in GBMs relative to metastases with a sensitivity of 80% and specificity of 90% [27]. Another group, Meier et al., used VASARI (Visually AcceSIble Rembrandt Images) and a support vector machine (SVM) to differentiate between GBMs and brain metastases. They found that VASARI MRI features of a well-defined non-enhancing tumor margin, ependymal disease, and tumor location successfully differentiated between GBMs and brain metastases [28].

The choice of contrast-enhanced T1-weighted pulse sequence can also improve the detection of intracranial or leptomeningeal metastases in general. Fu et al. demonstrated that PETRA (pointwise encoding time reduction with radial acquisition) was preferred for detecting osseous and leptomeningeal metastases, while DANTE-SPACE (delay alternating with nutation for tailored excitation sampling perfection with application-optimized contrasts using different flip angle evolution) was better than MPRAGE (magnetization-prepared rapid acquisition with gradient echo) and PETRA at demonstrating cerebral metastases [29]. This is likely because blood vessels are difficult to differentiate from small metastases on MPRAGE, and DANTE-SPACE can suppress the signal from vessels. Kim et al. showed that DANTE-SPACE improved both the time to detect and the accuracy of detecting enhancing metastases smaller than 5 mm [30]. Similarly, Danieli et al. determined that SPACE and VIBE (volumetric interpolated brain examination) produced superior conspicuity of brain tumors, including metastases, as compared to MPRAGE, and that SPACE displayed a more accurate tumor volume [31]. A downside of SPACE is the potential for an increased motion artifact; however, contrast-enhanced FLAIR MRI can be more helpful for detecting LC. A few studies have shown conspicuity of leptomeningeal disease was improved on delayed contrast-enhanced FLAIR images compared with contrast-enhanced T1-weighted images [32,33,34].

Interestingly, the primary origin of metastases may be inferred from the imaging appearance. For example, in a study of 952 brain metastases, SWI in combination with postcontrast T1-weighted imaging showed that hemorrhagic brain metastases were more likely to be melanoma (76.9% SWI positive) or breast cancer (55.6% SWI positive) [35]. Cao et al. used a radiomics model with binary logistic regression and an SVM to analyze both contrast-enhanced CT and T1-weighted MRI exams to differentiate between lung and breast cancer metastases [36]. Additionally, proton-MR spectroscopy (^1^H-MRS) and DSC or DCE MR perfusion can be helpful in further characterizing metastases. Huang et al. demonstrated that metastases from lung cancer were more likely than metastases from breast cancer or melanoma to have low choline to creatine (Cho/Cr) ratios on MRS. Metastases with a high Cho/Cr ratio were more likely to have a high normalized relative cerebral blood volume (CBV), suggesting that choline, a component of cell membranes and neurotransmitters, and vascular perfusion in neoplasms may be interrelated [37,38].

For patients with an identified primary breast cancer, subtypes also appear differently when metastasized to the brain. TNBC metastases are more likely to have cystic necrotic lesions identified on MRI Brain and are associated with more heterogeneity in T1W1 [39,40]. One research group was even able to use a linear regression model to determine that contour irregularity and a solid lesion composition could help differentiate HER2-positive from HER2-negative BCBM [41]. HER2-positive lesions correlate with a higher percent signal intensity change (PSIC) compared to HER2-negative lesions on MRI Brain, with PSIC demonstrating a sensitivity of 96% and specificity of 86% in differentiating HER2 status in 38 lesions [42]. Machine learning was also successful in detecting receptor conversion in breast cancer brain metastases. Luo et al. found that 51.3% of patients in their study had a discordance between the primary cancer receptors and the brain metastasis receptors, with loss of receptor expression more common than gain of expression. Their radiomics model was able to predict brain metastasis receptor status with accuracies of 78% for estrogen receptors, 83% for progesterone receptors, and 83% for HER2 receptors [43]. Cho et al. found that the primary breast cancer status and the brain metastasis receptor status were discordant in approximately 25% of cases. They were able to use machine learning classifiers to differentiate between (1) hormone receptor (HR)+/HER2−, (2) HR+/HER2+, (3) HR−/HER2+, and (4) triple-negative brain metastases with an accuracy of 90% [44]. Heitkamp et al. also used a random forest model with pre- and post-contrast T1-weighted and FLAIR images and patient age to predict the receptor status of breast cancer brain metastases. The AUC was 82% for ER+, 73% for PR+, and 74% for HER2+ [45].

Diagnosis of new metastatic lesions or progression of CNS disease can be difficult after radiation therapy or neurosurgical intervention. DSC perfusion is a useful tool in differentiating radiation necrosis from tumor-associated angiogenesis in MR, with a relative cerebral blood volume of 2.1 in DSC proving to be a reliable marker of tumor recurrence rather than necrotic changes [46]. While an increase in FLAIR signal after tumor resection has predicted local recurrence in the glioma literature, the same may be true for resection of brain metastases, with a specificity of 100% in a study of six cases, although with a poor sensitivity of about 50% [47]. Chemical exchange saturation transfer (CEST) MRI is a relatively new molecular imaging method that can depict amide groups upregulated in cancer cells and help differentiate treatment effects and the progression of disease after SRS [48,49]. It is gaining popularity in evaluating CNS metastases, with prior evidence for its use in evaluating gliomas [50,51,52].

In addition to detecting clinically significant and sizeable lesions seen in contrast-enhanced MRI studies, some imaging modalities under development map out a more extensive landscape of metastatic involvement for treatment planning. An animal model utilizing super-paramagnetic iron oxide nanoparticles (SPIONs) with near-infrared fluorescent dye/MR dual imaging demonstrated a much more detailed distribution of breast cancer brain metastases in vivo [53]. Another group from Canada has used bioluminescence imaging with firefly luciferase and a similar technology with cellular MRI in order to track cell viability in CNS metastasis in vivo [54].

### 3.2. Circulating Tumor Cells (CTCs)

Isolation of circulating tumor cells (CTCs) from peripheral blood or CSF is a type of liquid biopsy to identify and characterize metastatic disease in breast cancer. Several studies have suggested that CTC detection is more sensitive than standard CSF cytology for LM metastases and correlates with response to intrathecal chemotherapy and with later identification of positive cytology or MRI characterization of CNS disease [55,56,57,58]. CellSearch technology, which is currently FDA-approved to detect CTCs, demonstrated increased sensitivity for the detection of CTCs in CSF compared with cell cytology in a small prospective study of 40 breast cancer patients with suspected LC [56]. Studies have found that the sheer number of CTCs detected in peripheral blood is an independent prognostic factor for PFS and OS in patients with metastatic breast cancer, with a median OS of greater than 18 months in patients with fewer than five CTCs per sample but only about 6 months in those with greater than five CTCs [59,60,61]. Additionally, characterizing CTCs may have both diagnostic and prognostic implications. A small study of 26 metastatic breast cancer patients treated with immunotherapy demonstrated that those with peripheral blood CTCs expressing higher PD-L1 had longer OS after treatment with anti-PD-1 immunotherapy (*p* < 0.001) [62]. One study found that the detection of cytokeratin-19-expressing CTCs in peripheral blood is an independent risk factor for chemotherapy resistance among patients with early-stage breast cancer [63].

Application of CTC analysis to cohorts with BCBLM may have important prognostic implications, especially when evaluating response to therapy or characterizing the differential expression of tumor markers in metastatic lesions. Patel et al. were able to demonstrate that CTCs in CSF declined with systemic chemotherapy in BCBM patients, pointing to the possible use of CTCs in both diagnosis and surveillance of CNS disease [64]. Surveillance in peripheral blood is also important for patients with HER2-positive BCBM, as clearance of peripheral blood CTCs by day 21 of targeted therapy was found to correlate with 1-year OS in the LANDSCAPE trial [65]. While the HER2 status of metastases is presumed to be concordant with primary tissue biopsy, 38% (41/108) of a cohort of breast cancer patients with LC and a HER2-negative primary tumor were found to have HER2-positive CTCs in the CSF at our institution [66]. This raises an interesting question, as patients with HER2-positive CTCs may be eligible for HER2-directed therapy despite systemic HER2 negativity. Ultimately, targeted treatment of HER2-positive LC could improve survival and quality of life for BCBLM patients. However, clinical trials are needed for proof of this concept in this specific patient population.

Interestingly, one study identified a unique molecular signature of epithelial cell adhesion molecule (EpCAM)-positive CTCs in peripheral blood in patients with BCBM compared to those without CNS metastasis (Ki67+/uPAR+/int-B1+/NOTCH1+) [67]. Another was able to identify a population of EpCAM-negative CTCs isolated from the peripheral blood of breast cancer patients with a unique signature of HER2+/EGFR+/HPSE+/Notch1+ that was then able to generate brain metastases in nude mice as a xenograft, serving as a proof of concept [68]. Whether or not technologies looking at specific molecular signatures of CTCs can be utilized on a grand scale for clinical application remains to be seen. There is currently a clinical trial (NCT02941536) aimed at collecting CTCs before and after SRS and/or SFRT for BCBM patients to determine whether measuring the CTC number can be a reliable diagnostic and/or prognostic tool. Another trial funded by Biocept aimed to utilize a new proprietary lab test called “CNSide” to detect and quantify CTCs in CSF with greater sensitivity and in-depth analysis (NCT05414123).

### 3.3. Plasma Cell-Free DNA (cfDNA)

The study of cell-free tumor DNA from cerebrospinal fluid and blood has also gained interest for the diagnosis of brain and LM metastases. Circulating tumor DNA (ctDNA) has already been widely studied and adopted in clinical practice to assess disease response in peripheral blood, with proof of concept in reducing adjuvant chemotherapy use in colorectal cancer without compromising recurrence-free survival [69]. However, fewer studies have looked at ctDNA in CSF. In one small study, a machine learning model utilizing next generation sequencing (NGS) to analyze break-point motifs in CSF ctDNA was able to identify early LM metastases in patients with lung cancer [70]. Another study looking at CNS tumors and metastases demonstrated improved sensitivity for ctDNA in the CSF compared to peripheral blood plasma [71]. Finally, the ctDNA fraction in CSF was measured to be much higher in a small cohort of patients (*n* = 24) with breast cancer and LC compared to those without, and it had better sensitivity for detecting early LM disease compared to cytology and MRI [72].

### 3.4. Circulating miRNA, Extracellular Vesicles, and Proteins

A newer laboratory method involves isolating extracellular vesicles, i.e., exosomes or microvesicles, from peripheral blood in order to evaluate for biomarkers in metastatic breast cancer. The detectable number of small vesicles with specific protein cargo, including FAK and EGFR, have been found to be increased in patients with breast cancer compared to healthy controls, as have exosomal expression of splice variants of a pro-apoptotic protein called Survivin [73,74]. Assays that isolate and measure micro-RNA from these exosomes have demonstrated differential expression of certain micro-RNAs in patients with breast cancer, including miR-101, mIR-372, and miR-373, the latter of which is enriched in TNBC [75]. Interestingly, vesicle-derived miR-181c has been implicated in the breakdown of the blood–brain barrier with demonstrated in vivo brain uptake after systemic injection in mice [76]. A specific circular RNA found in plasma and exosomal compartments termed circBCBM1 has also been shown to promote the proliferation and migration of breast cancer cells to the brain in vivo and may prove diagnostic utility as a detectably circulating free miRNA in blood [77].

Detection of circulating proteins and metabolites in CSF is also of newer interest as a diagnostic and investigational method. Cancer cells have an altered metabolism compared to normal healthy cells, which could be leveraged in diagnostics similar to the way in which bacterial infections alter glucose and lactic acid concentrations in fluid compartments in the body. Metabolomic analysis of cerebrospinal fluid in patients with breast cancer and CNS involvement has identified 20 specific metabolites with differential upregulation compared to CSF collected from patients without malignant neoplasm, namely guanidine acetic acid, betaine, glucosamine/galactosamine, ornithine, methylcysteine, ethonalamine, aminophosphovaleric acid, 3-phosphoglycerate, 3PG and 2PG, 5-methyl-5-thioadenosine, cysteine, quinic acid, lactate, glutamic acid, 3-hydroxykyurenine, amino adipic acid, cystathionine, malic acid, succinate, and PEP [78]. There is also currently a clinical trial (NCT05286684) aimed at performing proteomic analysis of exosome material extracted from CSF.

## 4. Conclusions and Future Directions

While NCCN guidelines do not recommend screening asymptomatic breast cancer patients for CNS metastasis, studies have shown that BCBLM is likely much more prevalent than clinically recognized and may have implications for treatment. Early diagnosis of BCBLM can aid in earlier CNS-directed treatment and risk stratification, although this is likely to suffer from lead-time bias without a well-designed, randomized screening trial, as well as cost–benefit analysis to determine the utility of screening guidance such as in NSCLC. There are many new and exciting methods of diagnosis of CNS metastases in the field of neuro-oncology that are less invasive than brain biopsy, including advanced contrasted MR and ‘liquid biopsy’ of peripheral blood and CSF. Newer technologies that detect ctDNA and CTCs have already been incorporated in some clinical and research settings in prognostication, surveillance, and therapeutic decision-making.

## Figures and Tables

**Figure 1 cancers-16-03686-f001:**
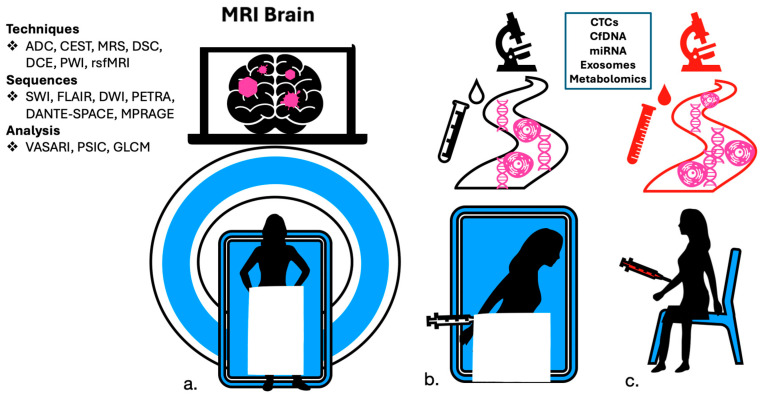
Methods in diagnosis and screening for brain and leptomeningeal metastases include (**a**) magnetic resonance imaging with contrast, (**b**) Cerebrospinal fluid (CSF) studies, and (**c**) peripheral blood studies.

**Table 1 cancers-16-03686-t001:** Selected Ongoing Trials in BCBLM Diagnostic Imaging (November 2023).

NCT Identifier	Official Title	Sponsor	Study Population	N	Status	Intervention	Outcome Measures	Results
NCT06072807	Brain [18F]-FES PET/CT in the Diagnosis, Treatment Planning and Response Assessment of Brain Metastases in Patients With Estrogen-Receptor Positive Breast Cancer	Weill Medical College of Cornell University	Patients with ER+ BCBM planned for radiation treatment	est: 20	not yet recruiting	18F- FES PET/CT scan in addition to standard imaging	Proportion of cases in which there is change in management based on FES PET/CT imaging	pending
NCT04246879	Diagnostic Accuracy of Delayed MRI Contrast Enhancement Characteristics and Radiation Necrosis Following Stereotactic Radiosurgery (SRS) for Brain Metastases	Duke University	Patients with brain metastases and signs of radiographic progression after stereotactic radiosurgery	est: 37	recruiting	Delayed MRI Contrast Enhancement	Detection of true tumor by delayed MRI as determined by surgical biopsy	pending
NCT05054998	A Pilot Study of Dual Time Point FDG PET MR Imaging Optimization for the Evaluation of Brain Metastasis	M.D. Anderson Cancer Center	Patients with brain metastases and plan for surgery or radiation, any solid organ metastases with at least 3 intraaxial and at least one enhancing >10 mm	est: 20	recruiting	FDG-PET/MRI	Optimal imaging time after radiotracer administration to maximize discrimination between lesions and healthy parenchyma	pending
NCT05911230	Advanced Diffusion MRI to Differentiate Tumor Recurrence From Pseudoprogression in Patients With Glioblastoma and Brain Metastases- AiD GLIO Pilot Trial	University Hospital, Basel, Switzerland	Patients with GBM or Brain metastases with suspected progression on standard MRI after first-line therapy and plan for surgical resection	est: 10	recruiting	ADW-MRI	Correlation between histopathology of resected tissue biopsy and tissue features from ADW-MRI	pending
NCT04111588	Diagnostic Assessment of Amino Acid PET/MRI in the Evaluation of Glioma and Brain Metastases	Norwegian University of Science and Technology	Patients with glioma or brain metastasis with planned surgery/SRS	est: 160	recruiting	Amino acid PET/MRI (18F-FACBC, 18F-FET, 11C-MET)	Differentiation of low and high-grade tumors at baseline, discrimination of recurrence vs. treatment-related changes	pending
NCT04244019	Differentiating Radionecrosis From Tumour Progression Using Hybrid FLT-PET/MRI in Patients With Brain Metastases Treated With Stereotactic Radiosurgery.	University Health Network, Toronto	Paitients with previously treated brain metastasis and new intracranial lesion suspicious for progression, undergoing planned resection	est: 30	recruiting	Hybrid FLT-PET/MRI	Differentiating radionecrosis from tumor; correlating with biopsy	pending
NCT05376878	Pilot Study to Evaluate 64Cu-DOTA-Trastuzumab Imaging in Patients With HER2+ Breast Cancer With Brain Metastatsis Treated With Fam-Trastuzumab Deruxtecan	City of Hope Medical Center	Patients with HER2+ BCBM, eligible with LMC	est: 10	recruiting	64Cu-DOTA-trastuzumab PET/MRI	Quantification of 64Cu-DOTA-trastuzumab uptake, comparison of SUVmax values in responders vs. non-responders, progression-free survival	pending
NCT04689048	Characterization of Large Brain Metastases With 18F-Fluciclovine PET/CT Treated With Staged Stereotactic Radiosurgery	Baptist Health South Florida	Patients with brain metastases with plan for SRS and at least one untreated lesion >2 cm	est: 20	recruiting	18 fluciclovine with PET/CT, MRI	Differences in sensitivity with 18F-fluciclovine enhancement in PET/CT and MRI, change in SUV parameters	pending
NCT03331601	Evaluation of 68GaNOTA-Anti-HER2 VHH1 Uptake in Brain Metastasis of Breast Carcinoma Patients	Universitair Ziekenhuis Brussel	Patients with BCBM, at least 1 lesion at least 8 mm diameter	est: 30	recruiting	68GaNOTA-Anti-HER2 VHH1 enhanced PET/CT	Tumor targeting potential using SUV values, change in uptake during or after treatment	pending
NCT05095766	Differentiation Between Radionecrosis and Tumor Recurrence for Post-stereotactic Radiosurgery Follow-up by Pharmacokinetic Analyses in Perfusion MRI and Positron Emission Tomography	Centre de recherche du Centre hospitalier universitaire de Sherbrooke	Patients with brain metastasis having unergone gamma knife radiosurgery and presenting for first MRI folow-up	40	active	DCE-MRI, FET PET	Differentiate radionecrosis from tumor recurrence with DCE-MRI, FET PET or combination imaging	pending
NCT04752267	Novel Dynamic PET Kinetics and MRI Radiomics Analyses in Brain Tumors	University of Southern California	Patients with primary brain tumor or metastatic tumors with documented radiation therapy	10	active	18F-FMAU PET/CT	Correlation between multiparametric MRI radiomics and dynamic 18F-FAMU PET to valuate tumors	pending
NCT00103038	NCI-Sponsored Multi-Disciplinary Study for MR Imaging of Intravenous Superparamagnetic Crystalline Particle Ferumoxytol in Primary High-Grade Brain Tumors and/or Cerebral Metastases	OHSU Knight Cancer Institute	Patients with high grade glioma, CNS lymphoma or brain metastases	155	completed	3T DCE-MRI with Ferumoxytol	Vascular Permeability (Ktrans), rCBV measurements, number and size of imaged metastases compared to gadolinium scans, overall survival	pending
NCT00938756	Interest of the Dosage of CA 15-3 in CSF for Diagnosing Carcinomatous Meningitis in Breast Cancer	Centre Oscar Lambret	Patients with breast cancer and LMC with or without BM, other malignancies with LMC included	est: 80	unknown	CA 15-3 levels in CSF	Measured levels of CA 15-3 in CSF of patients with LMC	pending
NCT03068520	Imaging After Stereotactic Radiosurgery for Brain Metastases or Primary Tumor Can Hybrid PET-MRI Differentiate Between Radiation Effects and Disease?	Assuta Medical Center	Patients with glial tumors or metastatic lesions treated with SRS	140	unknown	PET MR with 18F-DOPA	Distinguish between tumor and treatment related effects	pending

## Data Availability

No new data were created in the preparation of this manuscript, which is a review of the literature. The data presented in this review are referenced in the citations below.

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
