# Peer review of "Diagnostics and Screening in Breast Cancer with Brain and Leptomeningeal Metastasis: A Review of the Literature"

_cancers, 2024, doi:10.3390/cancers16213686_

Round 1

Reviewer 1 Report (Previous Reviewer 3)

Comments and Suggestions for Authors

I enjoyed the enhanced version of the manuscript, with screening strategies and a nice figure. Thank You.

Author Response

Comments 1: I enjoyed the enhanced version of the manuscript, with screening strategies and a nice figure. Thank You.

Response 1: We appreciate your input, thank you. 

Reviewer 2 Report (New Reviewer)

Comments and Suggestions for Authors

This is a review of the literature on screening and diagnosis of leptomeningeal metastasis. The review is well written and complete. There are descriptions of new and experimental techniques. The area of review is timely and important.  Aside from yellow highlighting that needs to be removed - not sure why this was added - I have very few concerns and/or questions.

Minor concerns/questions

1.  The authors state, " Interestingly, HER2-positive BC appears less 107 likely than luminal and TNBC to metastasize to the leptomeninges, which may have im-108 plications if screening for LC is ever advised in advanced breast cancer.18 "" Are these findings with or without the addition of adjuvant anti-HER2 therapy? Has the institution of HER2 therapy impacted the incidence of leptomeningeal metastasis?

2. There are many acronyms used throughout the manuscript. A table with a listing of the acronyms and their decoding would be helpful.

3. A table briefly describing the experimental techniques - including basis of analysis, contrast type if any, and strengths and weaknesses would be helpful.

Comments on the Quality of English Language

Minor English editing would be appreciated.

Author Response

Comment 1:  Aside from yellow highlighting that needs to be removed - not sure why this was added - I have very few concerns and/or questions.

Reply 1: We are required to submit revisions highlighted as per journal policy.

Comment 2: The authors state, " Interestingly, HER2-positive BC appears less 107 likely than luminal and TNBC to metastasize to the leptomeninges, which may have im-108 plications if screening for LC is ever advised in advanced breast cancer.18 "" Are these findings with or without the addition of adjuvant anti-HER2 therapy? Has the institution of HER2 therapy impacted the incidence of leptomeningeal metastasis?

Reply 2: Thank you for this comment. This statement summarizes demographic information from a retrospective review of patients diagnosed with LMD between 1997 and 2012 at MD Anderson. The manuscript does not include information on proportion of patients receiving HER2-directed therapies prior to LM diagnosis. While you raise an important point regarding the incidence of LMD after institution of HER2-directed therapies, the data makes this difficult to assess.

Comment 3: There are many acronyms used throughout the manuscript. A table with a listing of the acronyms and their decoding would be helpful.

Reply 3: Thank you for the suggestion. We have added a table to the manuscript, “Table 2: Acronym Glossary” to include common acronyms used in the manuscript.

Comment 4: A table briefly describing the experimental techniques - including basis of analysis, contrast type if any, and strengths and weaknesses would be helpful.

Reply 4: While this is a great suggestion, these methods are not available or provided in all reference manuscripts and thus we are unable to provide this table.

Reviewer 3 Report (New Reviewer)

Comments and Suggestions for Authors The authors have made an earnest attempt to address all the reviewers' comments.

Author Response

Comments 1: The authors have made an earnest attempt to address all the reviewers' comments.

Response 1: Thank you, we appreciate your input. 

This manuscript is a resubmission of an earlier submission. The following is a list of the peer review reports and author responses from that submission.

Round 1

Reviewer 1 Report

Comments and Suggestions for Authors

This study has the characteristic strengths and weaknesses of a literature review. 

There is only one table outlining ongoing studies. There should be at least one table addressing completed studies.

The paper does not provide a convincing argument of 1) why and 2) if yes, who should determine when and if screening should occur for these metastases. This is especially important since NCCN guidelines do not recommend screening in asymptomatic women

Reviewer 2 Report

Comments and Suggestions for Authors

This study reviewed some ongoing studies into screening and diagnostics for BCBLM as they relate to patient outcomes and prognostication. These studies include imaging methods such as MRI with novel contrast agents with or without PET/CT, as well as ‘liquid biopsy’ testing of the cerebrospinal fluid and serum to analyze circulating tumor cells, genomic material, proteins, and metabolites. Given recent advances in radiation, neurosurgery and systemic treatments for BCBLM, screening for CNS involvement should be considered in patients with advanced breast cancer as it may impact treatment decisions and overall survival.

1) I would recommend the authors to polish figure 1 to be more informative.

2) I recommend the authors to include some discussions on related studies using different omics data (PMID: 33461059; PMID: 35284940), which helps expand the scope of the study.

Comments on the Quality of English Language

The overall writing has some formatting issues, like wording, spacing, and some redundancy. I suggest the authors check the grammar and avoid any typos. More importantly, the writing needs improvement for readers to understand more easily.

Reviewer 3 Report

Comments and Suggestions for Authors

The review raises an important question of brain breast cancer metastasis, which is among the leading causes of death in these patients, and no screening strategy is currently approved.

I appreciate the presence of both Abstract and Short Summary.

The introduction clearly presents the current state of a problem in a laconic way and the purpose of the work, as well as methods, are clear. Further sections are also well and logically divided.

The "Diagnosis" section is ample, detailed, and also very interesting.

Tables and figures are informative and comprehensive. The citations are correct and adequate.

I have a few suggestions:
1. The authors' affiliations probably should be expanded, I was really curious where the authors were from.

2. line 40 - the "LM" abreviaton should be expanded at first use.

3. Line 128 - Please correct the word "imaging"

3. Clinical trials mentioned in the "Screening" section are great, could authors add some clinical trials ongoing for biomarkers research (for ex. TROP2) and few insights on genomic landscape. 

Thank You. I enjoyed reviewing.